# Experimental band structure spectroscopy along a synthetic dimension

Avik Dutt [1], Momchil Minkov[1], Qian Lin[2], Luqi Yuan [3], David A.B. Miller[1] & Shanhui Fan[1]

There has been significant recent interest in synthetic dimensions, where internal degrees of freedom of a particle are coupled to form higher-dimensional lattices in lower-dimensional physical structures. For these systems, the concept of band structure along the synthetic dimension plays a central role in their theoretical description. Here we provide a direct experimental measurement of the band structure along the synthetic dimension. By dynamically modulating a resonator at frequencies commensurate with its mode spacing, we create a periodically driven lattice of coupled modes in the frequency dimension. The strength and range of couplings can be dynamically reconfigured by changing the modulation amplitude and frequency. We show theoretically and demonstrate experimentally that time-resolved transmission measurements of this system provide a direct readout of its band structure. We also realize long-range coupling, gauge potentials and nonreciprocal bands by simply incorporating additional frequency drives, enabling great flexibility in band structure engineering.

[1] Ginzton Laboratory and Department of Electrical Engineering, Stanford University, Stanford, CA 94305, USA. [2] Department of Applied Physics, Stanford University, Stanford, CA 94305, USA. [3] School of Physics and Astronomy, Shanghai Jiao Tong University, Shanghai 200240, China. Correspondence and requests for materials should be addressed to L.Y. (email: yuanluqi@sjtu.edu.cn) or to S.F. (email: shanhui@stanford.edu)

The concept of band structure for periodic systems plays a central role in understanding the electronic properties of solid-state systems as well as the photonic properties of photonic crystals and metamaterials[1]. Recently, there has been significant interest in creating analogous periodic systems not in real space but in synthetic space, allowing one to explore higher-dimensional physics with a structure of fewer physical dimensions[2–12]. Synthetic dimensions are internal degrees of freedom of a system that can be configured into a lattice. For example, a synthetic dimension can be constructed from the equidistant frequency modes of a ring resonator. This synthetic frequency dimension enables one to study fundamental physics such as the effective gauge field and magnetic field for photons, 2D topological photonics in a 1D array, and 3D topological photonics in planar structures[8,9,13–18]. Moreover, the concept is interesting for applications such as unidirectional frequency translation, quantum information processing, nonreciprocal photon transport, and spectral shaping of light[19–30]. While most of the investigations of the frequency dimension have been theoretical, there have been a few recent experimental realizations[24,31]. Other realizations of synthetic lattices use the hyperfine spin states in cold atoms[4–6,32–36] or the orbital angular momentum (OAM) of photons[7,37–39]. In general, similar to standard solid-state and photonic structures, all these synthetic lattices are again characterized by a band structure in synthetic space, but a direct experimental measurement of this band structure in synthetic space is lacking.

In this work we provide a direct experimental demonstration of a band structure in the synthetic dimension. Specifically, we realize the synthetic dimension in a ring resonator containing an electro-optic modulator (EOM). By periodically driving the modulator at a frequency commensurate with the mode spacing or free-spectral range (FSR) of the ring, we introduce coupling between the modes and realize a synthetic frequency dimension lattice. The equidistance of the modes enables the realization of a long synthetic dimension with more than ten modes, all with uniform hopping implemented by a single modulation signal. Since the wavevector $k$ of the reciprocal space of such a frequency lattice is the time axis, we theoretically prove that temporally-resolved measurements of the transmission through the ring reveal its band structure, and demonstrate this method in experiments. Furthermore, we show that additional frequency drives enable us to engineer the band structure and to realize complex long-range coupling, photonic gauge potentials and nonreciprocal bands. We anticipate that the band-structure-measurement technique introduced here can be applied to a wide variety of geometries which utilize synthetic frequency dimensions, including those that show nontrivial topological physics[8,9,17,18,40–42].

## Results

**Theory.** We illustrate the concept that underlies the measurement of band structure in synthetic space using a simple model of a ring resonator. In the absence of group-velocity dispersion, the longitudinal modes of a ring resonator are equally spaced by the FSR, $\Omega_R/2\pi = c/n_g L$, where $n_g$ and $L$ are the group index and length of the ring respectively, and $c$ is the speed of light. In a static ring, these modes are uncoupled from each other. One can introduce coupling between the modes by incorporating a phase modulator in the ring (Fig. 1a). Here we consider on-resonance coupling where the modulation signal's periodicity $T_M = 2\pi/\Omega_M$ matches the roundtrip time of the ring, $T_R = 2\pi/\Omega_R$.

The equations of motion for the amplitude of the $m$-th mode $a_m$ for such a ring modulated at frequencies commensurate with

its FSR can be written as (see Supplementary Note 1),

$$\dot{a}_m(t) = im\Omega\, a_m + i\sum_n J_{mn}(t)\, a_n(t) \tag{1}$$

where $\dot{a}_m \equiv da_m/dt$, $\Omega_R = \Omega_M = \Omega$ and $T = 2\pi/\Omega$. $J_{mn}(t) = J_{mn}(t+T)$ is the coupling introduced by the periodic modulation signal $V_M(t)$. Throughout this paper, all frequencies are measured against the resonance frequency of the 0-th order mode. In Supplementary Note 1 we justify that $J_{mn}(t)$ depends only on $n-m$ and derive the explicit relation between $J_{n-m}(t)$ and $V_M(t)$ (Supplementary Eq. 30). By going to a rotating frame defined by $b_m = a_m e^{-im\Omega t}$, the equations of motion become,

$$i\dot{b}_m = -\sum_n J_{n-m}(t)\, b_n e^{i(n-m)\Omega t} \tag{2}$$

Defining a column vector $|b\rangle \equiv (\ldots, b_{m-1}, b_m, b_{m+1}, \ldots)^T = \sum_m b_m |m\rangle$, where $|m\rangle$ is the $m$-th unmodulated cavity mode, Eq. (2) can be written as a matrix equation

$$i|\dot{b}\rangle = H(t)|b\rangle. \tag{3}$$

Here $H(t)$ is the Hamiltonian with the matrix elements, $H_{mn}(t) = \langle m|H(t)|n\rangle = -J_{n-m}(t)e^{i(n-m)\Omega t}$.

This Hamiltonian $H(t)$ has two symmetries. The first is the modal translational symmetry along the frequency axis between the equally spaced modes, since the matrix element $H_{mn}$ depends only on $m-n$. This symmetry permits the definition of a conserved Bloch quasimomentum $k$ in the associated reciprocal space. Since the reciprocal space here is conjugate to the frequency dimension, we expect it to be identified with time. We will formally show that this is indeed the case below. The second symmetry is the time-translation symmetry $H(t) = H(t+T)$. This leads to Floquet bands[43,44] with quasienergies $\varepsilon_k$ that are defined in the interval $[-\Omega/2, \Omega/2]$. The relationship between the quasienergy $\varepsilon_k$ and the quasimomentum $k$ is the band structure.

Define the Bloch modes $|k\rangle = \sum_m e^{-im\Omega k}|m\rangle$. The state vector $|b\rangle$ can be written as $|b\rangle = (\Omega/2\pi)\int_{-\pi/\Omega}^{\pi/\Omega} dk\, \tilde{b}_k |k\rangle$. Equation (3) then reads,

$$i\dot{\tilde{b}}_k = \langle k|H(t)|b\rangle = H_k(t)\, \tilde{b}_k \tag{4}$$

$$H_k(t) = -\sum_s J_s(t)\, e^{is\Omega t - is\Omega k}; s \in \mathbb{Z} \tag{5}$$

where we have used $\langle k|H(t)|k'\rangle = \delta(k-k')H_k(t)$. $H(t)$ is already diagonal in $k$-space at each instant $t$ due to its modal translational symmetry. Since $H_k(t)$ is also time-periodic, the Floquet quasienergies $\varepsilon_{k,n}$ and eigenfunctions $\psi_{kn}(t)$ are well-defined and satisfy

$$(H_k(t) - i\partial_t)\psi_{kn}(t) = \varepsilon_{k,n}\psi_{kn}(t), \tag{6}$$

with $\psi_{kn} = \psi_{kn}(t+T)$, and $\varepsilon_{k,n} = \varepsilon_k + n\Omega$.

The above discussion was for a closed system. Next, we turn to an open system, where the ring is coupled to through- and drop-port waveguides (Fig. 2a), and show how its band structure can be read-out directly by time-resolved transmission measurements. Starting from Eq. (1), assuming all modes couple to both waveguides with equal rates $\gamma$, and by going to the rotating frame, the input-output equations are,

$$\dot{b}_m = -\gamma b_m + i\sum_s J_s(t)\, e^{is\Omega t} b_{m+s} + i\sqrt{\gamma}e^{-i(\omega + m\Omega)t} s_{in} \tag{7}$$

$$s_{out}(t) = i\sqrt{\gamma}\sum_m b_m(t)\, e^{im\Omega t} = i\sqrt{\gamma}\tilde{b}_k(t)\big|_{k=t} \tag{8}$$

where $s_{in}$ is the amplitude of the monochromatic input wave at

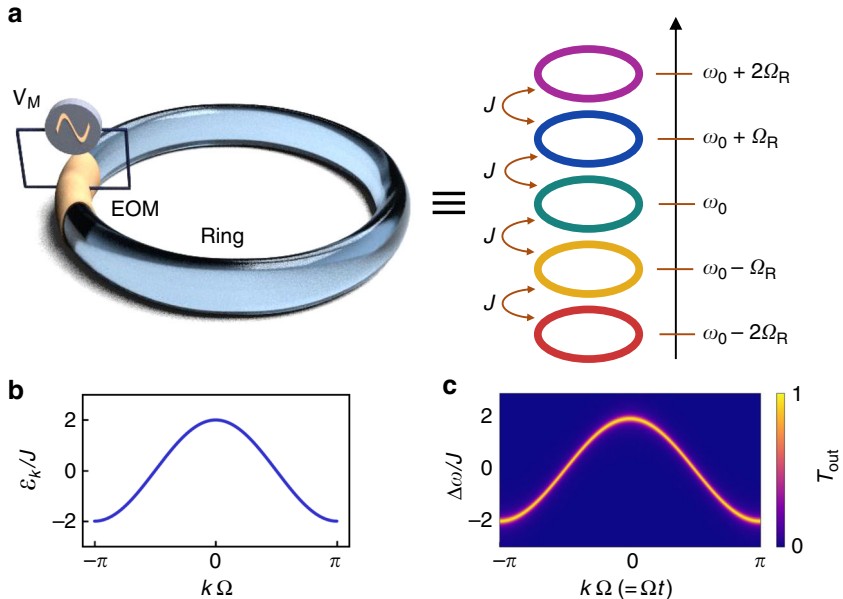

**Fig. 1** Dynamically modulated ring cavity and its band structure in the synthetic frequency dimension. **a** A sinusoidal signal $V_M$ modulates the index of a part of the ring (left) at the mode spacing $\Omega_R$, creating coupling $J$ between the different frequency modes (right). EOM electro-optic phase modulator. **b** Band structure of the nearest-neighbor coupled 1D tight-binding model, $\varepsilon_k = 2J\cos k\Omega$. **c** Theoretical time-resolved steady-state response of the ring (Eq. 16) for a cavity loss rate $2\gamma = J/5$. The time $t$ plays the role of the Bloch quasimomentum $k$

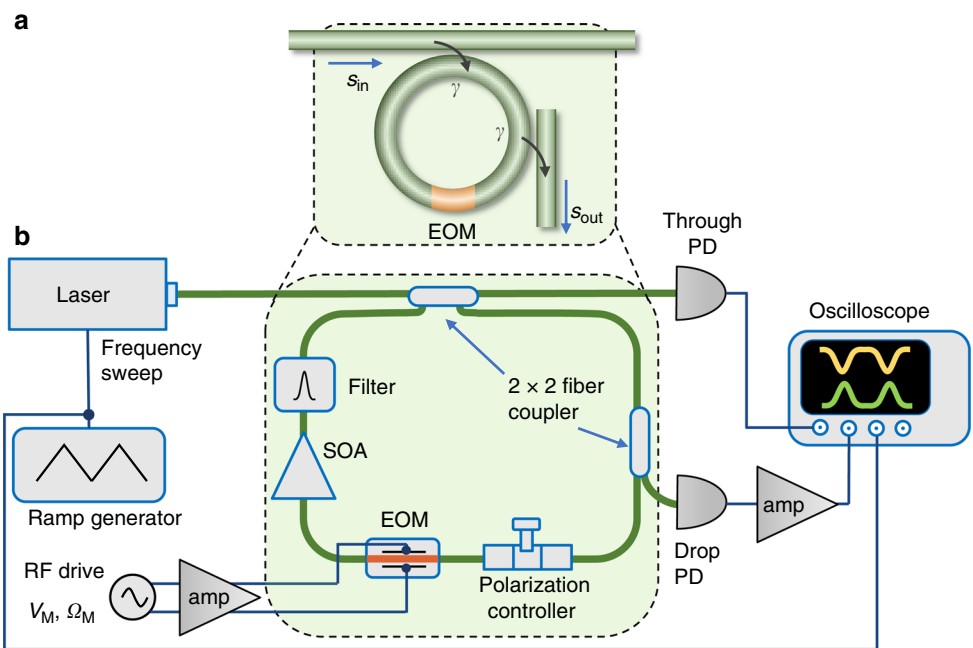

**Fig. 2** Experimental setup. **a** Schematic. **b** Detailed setup. Cavity length ~13.5 m, Cavity mode spacing or free-spectral range $\Omega_R = 2\pi \cdot 15.04$ MHz. EOM electro-optic phase modulator, PD photodiode, SOA semiconductor optical amplifier, amp RF amplifier

frequency $\omega$ (Fig. 2a). The last step in Eq. (8). follows from the definition of $\tilde{b}_k = \sum_m b_m e^{im\Omega k}$. It explicitly shows that the quasimomentum $k$ is mapped to the time $t$ in the cavity output field $s_{out}$. By defining a column vector $|s_{in}\rangle = s_{in} \sum_m e^{-im\Omega t}|m\rangle$, we can write Eq. (7) more compactly as:

$$i\partial_t|b\rangle = (-i\gamma + H(t))|b\rangle - \sqrt{\gamma}e^{-i\omega t}|s_{in}\rangle \quad (9)$$

At steady-state, we can write,

$$|b(t)\rangle = e^{-i\omega t}|b'(t)\rangle; \; |b'(t)\rangle = |b'(t+T)\rangle \quad (10)$$

From Eqs. (9) and (10) we have,

$$\langle k|[\omega - (H(t) - i\partial_t) + i\gamma]|b'\rangle = -\sqrt{\gamma}\langle k|s_{in}\rangle \quad (11)$$

or,

$$(\omega - (H_k(t) - i\partial_t) + i\gamma)\tilde{b}'_k = -\sqrt{\gamma}s_{in}T\delta(k-t) \quad (12)$$

Since $|b'\rangle$ is time periodic, the eigenstates $\psi_{kn}(t)$ of the Floquet Hamiltonian $H_k(t) - i\partial_t$ form a complete basis for expanding $\tilde{b}'_k$. These expansion coefficients can be obtained by taking the inner product of Eq. (12) with $\psi^*_{kn}(t)$, defined as $\langle f(t)|g(t)\rangle_T = (1/T)\int_0^T dt\, f^*(t) \cdot g(t)$:

$$\frac{1}{T}\int_0^T dt\, \psi^*_{kn}(t)(\omega - (H_k(t) - i\partial_t) + i\gamma)\tilde{b}'_k(t)$$
$$= -\sqrt{\gamma}s_{\text{in}}\int_0^T dt\, \psi^*_{kn}(t)\,\delta(k-t) \qquad (13)$$

Using Eq. (6) in Eq. (13), the inner product is,

$$\left\langle \psi_{kn}\Big|\tilde{b}'_k\right\rangle_T = -\frac{\sqrt{\gamma}s_{\text{in}}\,\psi^*_{kn}(t=k)}{\omega - \varepsilon_k - n\Omega + i\gamma} \qquad (14)$$

Finally, we can write the output field from Eq. (8) by using Eq. (10) and then expanding $\tilde{b}'_k(t)$ in the $\psi_{kn}$ basis,

$$s_{\text{out}}(t;\omega) = i\sqrt{\gamma}e^{-i\omega t}\sum_n \psi_{kn}(t)\langle\psi_{kn}|\tilde{b}'_k\rangle_T\big|_{k=t}$$
$$= -e^{-i\omega t}s_{\text{in}}\sum_n \frac{i\gamma\,|\psi_{kn}(t)|^2}{\omega - \varepsilon_k - n\Omega + i\gamma}\bigg|_{k=t} \qquad (15)$$

Equation (15) shows that the transmission at time $t$ is exclusively determined by the quasienergies and eigenstates at $k = t$. For $\gamma \ll \Omega$ and $|J_{n-m}| < \Omega/2$, only the term for which $n\Omega$ is closest to the input frequency $\omega$ contributes significantly to the sum in Eq. (15). Using this $n$, and denoting the input detuning by $\Delta\omega \equiv \omega - n\Omega$, we can write the normalized transmission $T_{\text{out}} = |s_{\text{out}}/s_{\text{in}}|^2$ as,

$$T_{\text{out}}(t=k;\Delta\omega) = \frac{\gamma^2}{(\Delta\omega - \varepsilon_k)^2 + \gamma^2}|\psi_{kn}(t)|^4 \qquad (16)$$

Equation (16) shows that for a fixed input detuning $\Delta\omega$ that is within a band of the system, the temporally-resolved transmission exhibits peaks at those times $t$ for which the system has an eigenstate with $\varepsilon_k = \Delta\omega$, $k = t$. Thus, measuring the times at which the transmission peaks appear in each modulation period $2\pi/\Omega$, as a function of $\Delta\omega$, yields the Floquet band structure of the system. This is in contrast to previous proposals of detecting density of states in real-space Floquet systems[45,46]. These proposals do not directly reveal the $k$-dependence of eigenenergies, and hence do not provide a direct detection of the band structure.

When the magnitude of $J$ is much smaller than $\Omega$, one can use the rotating wave approximation in Eq. (2), by Fourier expanding $J_{n-m}(t)$ and keeping only the terms on the right-hand side of Eq. (2) that are time-independent. This allows us to define an effective time-independent Hamiltonian, $H_k^{\text{eff}} = -\sum_s \tilde{J}_{s;q=-s}\,e^{-is\Omega k}$, where $\tilde{J}_{s;q} \equiv (1/T)\int_0^T dt\, J_s(t)e^{-iq\Omega t}$. As an example, suppose $J_s(t) = -2J_1\cos\Omega t$, then the system has the band structure of a 1D nearest-neighbor-coupled tight-binding model, $\varepsilon_k = 2J_1\cos k\Omega$ (Fig. 1b). In Fig. 1c we plot the numerically calculated time-resolved transmission of Eq. (16) obtained by diagonalizing the full Floquet Hamiltonian without making the rotating wave approximation, which agrees well with the band structure in Fig. 1b. For details of the numerical diagonalization, see "Methods".

**Experimental setup**. We implement the synthetic frequency dimension using a fiber ring resonator incorporating an electro-optic lithium niobate phase modulator, as shown in Fig. 2b. The ring has a roundtrip length of ~3.5 m, corresponding to a mode spacing $\Omega_R = 2\pi \cdot 15.04$ MHz (see "Supplementary Methods")[47]. We use a narrow linewidth continuous wave (cw) laser as the input. Its frequency could be scanned by a range much larger than $\Omega_R$ to observe multiple Floquet bands beyond the first Floquet Brillouin zone. The setup also includes a semiconductor

optical amplifier (SOA) to partially compensate various losses, including the loss from the modulator. The residual loss and the input coupling leads to a cavity photon decay rate of $2\gamma = 2\pi \cdot 300$ kHz. The setup is stable for more than 1 ms, which is sufficient for obtaining the entire band structure. Thus there is no need for active feedback stabilization. Note that Spreeuw et al. have reported band gaps in a Sagnac fiber ring using Faraday elements and counterpropagating modes; however, the absence of frequency-dimension coupling precluded the mapping out of the entire band structure[48].

To measure the time-resolved transmission that is necessary to read out the band structure, we monitor the through- and drop-port outputs on a fast photodiode (bandwidth > 5 GHz), connected to a 1 GHz oscilloscope. We scan the laser frequency slowly at 100–500 Hz such that the system reaches steady state at each frequency. This enables us to map out the band structure in a line-by-line raster scan fashion.

**Experimental results**. We plot the experimentally measured band structure in Fig. 3a–c, where the modulation voltage has a form $V_M(t) = V_1\cos\Omega t$, and observe excellent agreement with the theoretically calculated band structure for a nearest-neighbor coupled 1D lattice based on Eq. (15) (Fig. 3d–f). Both the cosine dependence of the band on the quasimomentum $k(=t)$, and the increase of the width of the band with increasing modulation amplitude are observed. At a fixed detuning $\Delta\omega$, the transmission response of the system is $2\pi/\Omega_M$-periodic along the time axis. The response is also periodic along the $\Delta\omega$-axis. Both of these periodic responses are expected due to the modal translational symmetry and time periodicity as discussed earlier.

The results in Fig. 3a–c were obtained using a fast photodiode with a bandwidth greater than 5 GHz. If we instead use a slower photodiode with a bandwidth less than the modulation frequency $\Omega_M$, the photodiode provides a time-averaged response, and we observe transmission spectra as shown in Fig. 3g–i. Such transmission spectra represent a direct observation of the photonic density of states (DOS) of the synthetic-space lattice, as can be seen by integrating Eq. (15) over $k$, which yields the imaginary part of the Green's function for the band, and hence the DOS, in the limit of weak waveguide-cavity coupling ($\gamma \to 0$), and assuming $|\psi_{kn}(t)|$ to be independent of $k$. As a demonstration, the red dashed lines in Fig. 3h denote the DOS of a 1D lattice with nearest-neighbor coupling, and match the experimentally measured data well after accounting for the smearing due to a finite $\gamma$. The van Hove singularities associated with the DOS of periodic systems are also clearly visible at the edges of the band[49–51].

In the synthetic space, it is straightforward to create a wide variety of band structures by simply changing the modulation pattern. Different modulation patterns correspond to different coupling configurations in the tight-binding lattice[52]. Such a flexibility is unique to synthetic space and is unmatched in either solid-state materials or photonic crystals. As an illustration, long-range coupling can be achieved by using a modulation with a frequency that is a multiple of the FSR[31,52]. Figure 4a shows the measured band structure of the system when $\Omega_M = 2\Omega_R = 2\pi \cdot 30.08$ MHz, which creates a lattice with only next-nearest neighbor coupling. This system has a response that is periodic at a frequency of $2\Omega_R$. Thus, the first Brillouin zone extends from $k = -\pi/2\Omega_R$ to $\pi/2\Omega_R$, which is half the extent shown in Fig. 4a. The resulting measurement shown in Fig. 4a agrees with the band structure for a tight-binding model with only next-nearest neighbor coupling.

Moreover, the inclusion of both nearest-neighbor and long-range hopping leads to a photonic gauge potential whose effects can be observed in the band structure[52]. As a demonstration, we

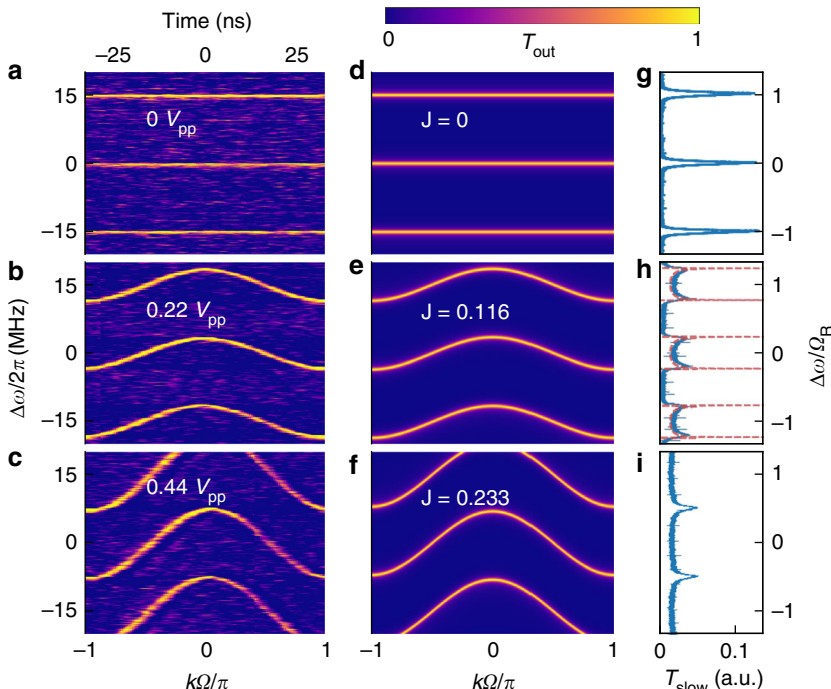

**Fig. 3** Band structure for on-resonance modulation $\Omega_M = \Omega_R = 2\pi \cdot 15.04$ MHz. **a–c** Experimentally measured time-resolved transmission. **d–f** Theoretical transmission based on Eq. (15). The bottom x-axis represents time (the quasimomentum) in units of $\pi/\Omega_R$, while the top x-axis of (**a–c**) presents the time in real units. The y-axis represents the input laser's detuning from the ring's resonance frequency, allowing us access to different quasienergies $\varepsilon_k$ (left: real units, right: normalized to FSR). On increasing the coupling $J$ (theory) or the amplitude of modulation $V_{pp}$ (experiment), cosinusoidal bands with increasing extent in quasienergy are observed, in agreement with $\varepsilon_k = 2J \cos k\Omega$. See also Supplementary Movie 1. **g–i** Corresponding time-averaged transmission through the modulated cavity. In (**g**), typical Lorentzian resonances of the unmodulated cavity are seen. In (**h**), the photonic density of states (DOS) in synthetic space is observed. The experimental data (blue) matches the expected density of states (red dashed) of a 1D tight-binding model with nearest neighbor coupling. In the bottom panels (**c**), (**f**, **i**), the modulation-induced coupling is strong enough to cause overlap of Floquet bands

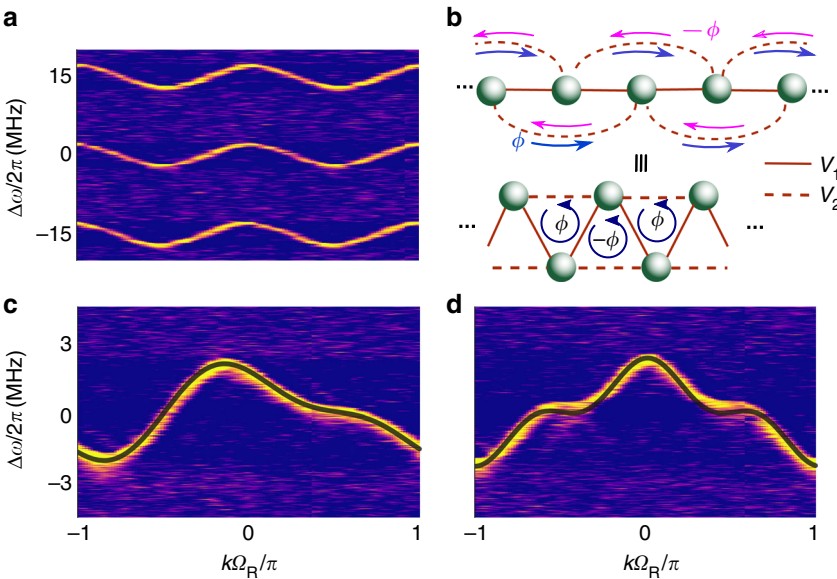

**Fig. 4** Band-structure engineering with long-range hopping and synthetic gauge potentials. **a** Time-resolved transmission for a modulation frequency $\Omega_M = 2\Omega_R$, coupling exclusively next-nearest neighbors. While the band structure is similar to Fig. 3, the first Brillouin zone is halved, and two periods of the cosine band structure are seen in $\Omega_R t \in [-\pi, \pi]$. **b** Schematic of lattice for $V_M = V_1 \cos \Omega_R t + V_2 \cos(n\Omega_R t + \phi)$, where $V_1$ (solid lines) determines the strength of nearest-neighbor coupling and $V_2$ (dashed lines) determines the long-range hopping, here shown for $n = 2$. The phase of the $V_2$ links is $+\phi$ when going up in frequency and $-\phi$ when going down in frequency, as imposed by the modulation[53,54]. Bottom: Equivalent lattice representation by a triangular chain threaded by a phase $\phi$ per plaquette. **c** Measured nonreciprocal band structure for $V_2/V_1 = 0.40$, $\phi = \pi/2$, $n = 2$, which shows strong asymmetry about $k = 0$. See also Supplementary Movie 2. **d** Same as **c**, but for $\phi = 0$, $n = 3$. Black overlays in (**c**) and (**d**) indicate expected band structures

apply a modulation signal of the form $V_M = V_1 \cos \Omega_R t + V_2 \cos (2\Omega_R t + \phi)$. In this case, $\phi$ is the photonic gauge potential, as can be seen by representing the corresponding tight-binding lattice in terms of a collection of plaquettes, and by noticing that $\phi$ corresponds to a magnetic flux that threads each plaquette (Fig. 4b)[53–55]. Figure 4c shows the experimentally obtained band structure for $\phi = \pi/2$. Note that this band is asymmetric around $k = 0$, and hence nonreciprocal. This indicates the breaking of time-reversal symmetry in the structure due to the presence of the gauge potential $\phi$. In Fig. 4d we show the band structure for an even longer range hopping, obtained by applying a modulation signal $V_M(t) = V_1 \cos \Omega t + V_2 \cos 3\Omega t$. The range of coupling that we can achieve is limited to third-nearest neighbor coupling by the 50-MHz analog bandwidth of the arbitrary waveform generator (AWG) used in our setup (see "Supplementary Methods"). With the use of AWGs with much higher analog bandwidths exceeding 1 GHz, which are commercially available, it should be straightforward to significantly extend the range of coupling. Alternatively, one could use larger rings with a smaller FSR ~5 MHz, which would permit up to tenth-nearest neighbor coupling within a 50 MHz bandwidth of the AWG.

## Discussion

We have theoretically proposed and experimentally demonstrated a technique to directly measure the band structure of a system with a synthetic dimension. The fiber ring resonator with a modulator allows for independent tuning of the strength and range of the coupling along this synthetic lattice, making it dynamically tunable. By combining multiple frequency drives and incorporating long-range hopping, we have demonstrated a photonic gauge potential and its effect on the band structure.

The synthetic frequency dimension platform that we have experimentally demonstrated here, along with the band structure measurement technique, is ripe for probing systems beyond 1D[10–12,18,56–58,60]. For example, 2D quantum Hall phenomena such as one-way edge states[8,9,61] and synthetic Hall ribbons[5,34,62–65] could be observed in extensions of our system, with the added benefit of frequency conversion from transport along the synthetic dimension. The dimensionality can be increased beyond 1D by using real-space dimensions[8,9], by using additional frequency dimensions[26,52], by using the Floquet dimension[14,59,60] or by using other synthetic dimensions such as OAM in conjunction with frequency[66] (see Supplementary Discussion). Even within 1D, there have been proposals to realize unique photon transport phenomena using dynamically modulated cavities, which could be implemented in a reconfigurable fashion in our platform[16,67]. Longer fiber ring resonators in the pulsed regime have been previously used for realizing parity-time symmetry[68], optical Ising machines[69] and soliton interactions[70], in a synthetic temporal dimension[71–74] that is complementary to our cw-pumped synthetic frequency dimension. In these systems, the band structure has been indirectly inferred from transport measurements in the synthetic temporal dimension, using pulses in fiber loops to simulate photonic lattices[75]. Lastly, the advent of on-chip silicon[76,77] and lithium niobate ring resonators[78] with modulation bandwidths higher than the FSR of on-chip ring resonators can enable synthetic dimensions and topological photonics in a monolithically integrated platform.

## Methods

**Numerical diagonalization of the Floquet Hamiltonian.** In this section we outline the procedure we used to diagonalize the Floquet Hamiltonian from Eq. (5) and numerically calculate the quasienergies and eigenfunctions.

The Floquet eigenstates at each $k$ are time periodic and can be expanded in terms of their Fourier components,

$$\psi_{kn}(t) = \sum_p e^{ip\Omega t} \tilde{\psi}_{kn}(p) = \sum_p |p\rangle \langle p|\psi_{kn}(t)\rangle_T \qquad (17)$$

with the basis vectors $|p\rangle = e^{ip\Omega t}$ and the inner product

$$\langle f(t)|g(t)\rangle_T = \frac{1}{T}\int_0^T dt\, f^*(t) \cdot g(t). \qquad (18)$$

In this basis, the matrix elements of the Floquet Hamiltonian are,

$$\langle p'|H(t) - i\partial_t|p\rangle_T = \frac{1}{T}\int_0^T dt\, e^{i(p-p')\Omega t}(H_k(t) + p\Omega) \qquad (19)$$

$$= p\Omega\delta_{pp'} - \sum_s e^{-is\Omega k}\tilde{J}_{s;p'-p-s} \qquad (20)$$

where $\tilde{J}_{s;q} \equiv (1/T)\int_0^T dt\, J_s(t)e^{-iq\Omega t}$ is the $q$-th Fourier component of the periodic hopping term $J_s(t)$ between modes $m$ and $m+s$ ($q \in \mathbb{Z}$). To calculate $\psi_{kn}(t)$ and $\varepsilon_{k,n}$, we truncate the matrix corresponding to the Floquet Hamiltonian in the $|p\rangle$ basis to a large enough order $p_{max}$ and numerically determine the eigenvalues and eigenvectors, respectively[67,79].

**Experimental calibration of the frequency axis.** The laser's optical frequency can be scanned by applying a voltage to the frequency sweep input. To calibrate this relationship, we could apply a sinusoidal modulation to the intracavtiy EOM while simultaneously scanning the laser's frequency with a linear voltage ramp. The sidebands created by the modulation provide a calibration of the frequency change with voltage. A more accurate calibration can be obtained by varying the modulation frequency $\Omega_M$ till the slow transmission is maximally flattened for moderate modulation amplitudes, $J \approx 0.1\,\Omega$, as in the observation of the DOS in Fig. 3h. Multiple resonances of the cavity were simultaneously flattened for $\Omega_M$ close to $2\pi \times 15.04$ MHz, indicating that the resonances are equally spaced by this FSR. Since we applied a linear voltage ramp signal, we could equivalently calibrate the frequency axis with time. The relationship between voltage and frequency was found to deviate from linearity for large frequency sweeps, as shown in Supplementary Fig. 1. We observed that a quadratic fit was of sufficient accuracy to relate the frequency with the scan time over an optical frequency scanning range of 60 MHz. This fit was then used to convert the slow time axis from the raw data into the frequency axis in Figs. 3 and 4.

**Data acquisition and time slicing.** Each band structure measurement consisted of a 1-ms acquisition of the drop-port transmission through the ring cavity. The drop-port output was sent to an erbium-doped fiber amplifier (EDFA) to boost the signal-to-noise ratio. The EDFA output was detected on a 5 GHz photodiode, whose output was further electrically amplified by an RF amplifier (passband 50 kHz to 14 GHz) before being sent to the 1 GHz oscilloscope. A finite lower cutoff frequency of the RF amplifier mitigates the added spontaneous emission noise from the EDFA and the intracavity SOA. We used a sampling rate of 2 GSa/s to get sufficient resolution along the (fast) time axis corresponding to the Bloch quasi-momentum $k$ in Figs. 3 and 4.

The 1-ms-long trace was then broken into time slices, each of duration equal to the roundtrip time of the ring (=1/(15.04 MHz) ≈66.49 ns). The time within each of these slices (the "fast" time) corresponds to $k$. The change of the starting time between successive slices was converted to a change in optical frequency of the laser using the calibration in Supplementary Fig. 1. Each such slice shows zero, one or more peaks depending on whether the input laser frequency is outside any band, at the edge of a band, or within a band, respectively [see Supplementary Movies 1 and 2]. Stacking up the slices vertically yields the band structure in Figs. 3a–c and 4a, c and d.

## Data availability

The data that support the findings of this study are available from the corresponding author on reasonable request.

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

## Acknowledgements

This work is supported by a Vannevar Bush Faculty Fellowship (Grant No. N00014-17-1-3030) from the U. S. Department of Defense, and by a MURI grant from the U. S. Air Force Office of Scientific Research (Grant No. FA9550-17-1-0002). M.M. acknowledges support from the Swiss National Science Foundation (Grant No. P300P2_177721).

## Author contributions

Q.L. conceived the band structure measurement technique. A.D. designed, built, and characterized the setup, in consultation with L.Y., Q.L., D.A.B.M., and S.F. M.M. developed the coupled-amplitude equations and the Floquet analysis relating the transmission to the band structure. A.D. collected and analyzed the experimental data. All authors contributed to discussion of the results and writing the manuscript. S.F. supervised the project.

## Additional information

**Competing interests:** The authors declare no competing interests.

