## [Peer Review File · Nature Communications]

Reviewers' comments:

Reviewer #1 (Remarks to the Author):

The paper reports the design of a dynamically modulated ring resonator system to create a synthetic frequency dimension lattice. Experimentally they measured the corresponding band structure along the synthetic frequency dimension by time-resolved transmission measurements. Additionally, they have demonstrated different gauge potentials and its implications by inducing long range hopping in the synthetic lattice. This work gives a comprehensive study of synthetic space lattice with corresponding band structure picture. This is a very timely work since only until very recently there has been interesting works on synthetic dimensions and this work adds nicely to the experimental efforts.

Overall, the paper is well organized in terms of theory and follows up with the corresponding experiment. The theoretical claims were addressed and explained via reasonable experimental facts. I am happy to recommend the publication of this work in Nature Communications once the following points regarding the outlooks are addressed.

Comments:

1. The use of SOA in a looped geometry will certainly enhance the background photon flux. So could the authors add some thoughts on the limitations of such setup in terms loss and how that will play a role in the quality of band structure? Also are there any tricks the authors have used to circumvent these issues? See comment 3.
2. It is impressive to see the next nearest interaction by tuning the modulation frequency. So can the authors comment on if one can look at even the higher order long range interactions on this synthetic dimension?
3. What are the main challenges one will face when trying to extend this design to a higher dimensional lattice? And how can one overcome such issues?

Reviewer #2 (Remarks to the Author):

In their manuscript "Experimental band structure spectroscopy along a synthetic dimension," Dutt et al. theoretically proposed and experimentally demonstrated a technique to directly measure the band structure of a system with a synthetic dimension. To this end, they implemented a fiber ring resonator with a modulator that allows for independent tuning of the strength and range of the coupling along a synthetic lattice.

This is a very nice paper, I like it a lot. The idea is clearly presented, the experiments are convincing, the told story is smooth and interesting. I searched for things to criticize in the manuscript, I did not find any. There may be one optional thing: They authors might want to cite a recent arXiv about synthetic dimensions (arXiv: 1903.07883); but this is completely up to the authors.

I recommend publication of the manuscript in Nature Communications.

Reviewer #3 (Remarks to the Author):

In this manuscript, the authors present the paradigm of generating a synthetic dimension using parametric coupling between the free-spectral range states of a ring resonator. The scheme is, then, successfully realized experimentally. The field of generating synthetic dimensions for quantum simulation is nowadays extremely active. In this context, the authors provide an important but technical stepping stone towards the realization of more interesting physics. At the same time, the authors present a nice comprehensive theoretical overview of the Floquet analysis for the detection of the synthetic band spectrum. Combining all of this together, I however do not

see the added merit of publishing this work in Nat. Comm. due to the following reasons:

1. The theoretical analysis is not new:

- a. The proposal of using this system for synthetic dimensions has already been proposed by some of the authors of this paper, as well as in other works [9, 39, Nature Physics 13, 545 (2017)].
- b. The input-output formalism for detection of photonic bands has been around for some time, see e.g., [Phys. Rev. A 84 043804 (2011), Phys. Rev. Lett. 112 133902 (2014)].
- c. The time analysis for reading of Floquet bands is also widely used, for example, [Phys. Rev. X 4, 031027 (2014), Phys. Rev. A 96, 053602 (2017)].

2. Recently, there have appeared more advanced results on photonic realization of synthetic dimensions [Nature 567, pages356–360 (2019)].

We thank all the reviewers for their comments and suggestions. Below we report our responses in green and indicate the modifications we have incorporated in our manuscript to address them in red. All page numbers and reference numbers are based on the revised manuscript, unless specifically mentioned otherwise.

Reviewer #1 (Remarks to the Author):

The paper reports the design of a dynamically modulated ring resonator system to create a synthetic frequency dimension lattice. Experimentally they measured the corresponding band structure along the synthetic frequency dimension by time-resolved transmission measurements. Additionally, they have demonstrated different gauge potentials and its implications by inducing long range hopping in the synthetic lattice. This work gives a comprehensive study of synthetic space lattice with corresponding band structure picture. This is a very timely work since only until very recently there has been interesting works on synthetic dimensions and this work adds nicely to the experimental efforts.

Overall, the paper is well organized in terms of theory and follows up with the corresponding experiment. The theoretical claims were addressed and explained via reasonable experimental facts. I am happy to recommend the publication of this work in Nature Communications once the following points regarding the outlooks are addressed.

We thank the reviewer for his/her comments recognizing the timeliness and significance of our work.

Comments:

1. The use of SOA in a looped geometry will certainly enhance the background photon flux. So could the authors add some thoughts on the limitations of such setup in terms loss and how that will play a role in the quality of band structure? Also are there any tricks the authors have used to circumvent these issues? See comment 3.

We add the following details to the Supplementary Methods to address the reviewer's question:

"The SOA in the cavity increases the background photon flux that is not associated with the band structure. In our case the background photon flux does not significantly hinder the band structure measurement, since we control the SOA gain to operate the cavity sufficiently below the lasing threshold such that the peaks in the time-resolved transmission measurement can be located correctly. Some of the background noise seen outside the bands in Figs. 3 and 4 is attributable to the SOA. While the SOA helps to compensate the EOM's insertion loss and achieve a high $Q > 600$ million, the maximum achievable drop-port transmission efficiency was observed to be still limited to 5% due to intracavity losses. Attempts to increase the efficiency by increasing the SOA gain initiates lasing.

The issues discussed above are specific to our fiber-based implementation due to the substantial insertion loss of the EOM. The implementation of synthetic frequency dimension in cavity systems

with low-insertion loss modulators, such as integrated lithium niobate rings, may not need an intracavity optical amplifier and hence may not suffer from the issues of background photon flux.”

Regarding tricks to circumvent these issues in our setup, we used an RF amplifier after the fast photodiode to increase the signal-to-noise ratio. Since the RF amplifier was AC coupled with a low frequency cutoff of 50 kHz, the background DC flux was automatically filtered out in the detection. We clarify this in the Methods section on “Data acquisition and time slicing” by incorporating the following modifications:

“The EDFA output was detected on a 5 GHz photodiode, whose output was further electrically amplified by an RF amplifier (passband 50 kHz to 14 GHz) before being sent to the 1 GHz oscilloscope. A finite lower cutoff frequency of the RF amplifier mitigates the added DC background photon flux from the EDFA and the intracavity SOA.”

Furthermore, in Supplementary Methods, we have already described the role of the intracavity optical bandpass filter in suppressing ASE from the SOA: “A dense wavelength-division multiplexing (DWDM) filter with a center wavelength of 1542.14 nm (Channel 44) and a 26.5 GHz passband was used after the cavity for two reasons. First, it prevented spurious lasing near the peak gain wavelength of the SOA, or at wavelengths where the roundtrip loss of the cavity is lower than at the input laser wavelength. Second, it helped to filter out amplified spontaneous emission noise outside the passband of the filter.”

2. It is impressive to see the next nearest interaction by tuning the modulation frequency. So can the authors comment on if one can look at even the higher order long range interactions on this synthetic dimension?

We have demonstrated both a next-nearest neighbor coupling and a third-nearest neighbor coupling [see Fig. 4(d)]. It is readily possible to extend this to a much higher order long-range coupling by increasing the modulation frequency. In our experiment, we used an arbitrary waveform generator with an analog bandwidth of 50 MHz to generate the modulation signals. This restricted us to third order coupling (45 MHz modulation) for the 15 MHz FSR ring in our setup. Arbitrary waveform generators with analog bandwidths of a GHz or more are commercially available. Alternatively, one could use rings with a smaller FSR to permit an even longer range coupling without increasing the required analog bandwidth. We summarize these comments by adding the following sentences at the end of the Results section:

“The range of coupling that we can achieve is limited to third-nearest neighbor coupling by the 50-MHz analog bandwidth of the arbitrary waveform generator (AWG) used in our setup (see Supplementary Methods). With the use of AWGs with much higher analog bandwidths exceeding 1 GHz, which are commercially available, it should be straightforward to significantly extend the range of coupling. Alternatively, one could use larger rings with a smaller FSR \sim 5 MHz, which would permit up to tenth-nearest neighbor coupling within a 50 MHz bandwidth of the AWG.”

Additionally, we add a paragraph to the end of the Supplementary Discussion [see response to next comment 3] discussing the limits placed by dispersion and the finite bandwidth of cavity components on the number of modes that can be coupled.

3. What are the main challenges one will face when trying to extend this design to a higher dimensional lattice? And how can one overcome such issues?

There are several avenues one could pursue to extend this design to a higher dimensional lattice: using real-space dimensions [Refs. 8,9], using non-frequency synthetic dimensions in conjunction with frequency (Ref. [66]), and using additional frequency dimensions [Ref. 53, 34]. Since theoretical proposals exist for all these schemes, we focus on the experimental challenges here.

First, one could extend the lattice along a real spatial dimension, as in Refs 8,9, by evanescently coupling multiple rings. The main challenge here would be to match the resonance frequencies of all the rings in the lattice, which can be overcome by actively locking the lengths of each ring using a feedback stabilization scheme.

Second, one could combine other internal degrees of freedom, such as the orbital angular momentum (OAM) of light, and form a lattice with one frequency dimension and one OAM dimension. Preliminary experiments along this direction have been reported in Ref. 20, where Cheng et al. demonstrated a resonator with 46 nearly degenerate Laguerre-Gauss modes. Further work is needed to couple these modes along the OAM dimension and the frequency dimension by incorporating spatial-light modulators and electro-optic phase modulators, respectively, within the resonator.

Third, one could use the Floquet dimension generated by time-periodic modulation, independent of the FSR-separated modes, as in Refs. 22 and 60. This synthetic lattice, however, is concomitant with a synthetic electric field along the Floquet dimension, so one would have to include its effects in the theoretical description and experimental measurements.

Finally, one could use the strategies outlined in Refs. 53 and 34 to add modulations at n times the FSR ($n \gg 1$), to form an effective 2D or higher dimensional lattice, still within a single resonator. Experimentally, this is the most straightforward of the three schemes, but it is somewhat restrictive in terms of the lattice geometries that can be achieved. For example, it would involve spurious long-range coupling between the 1D strips of a 2D square lattice, and it is difficult to achieve arbitrary gauge potentials in such a system. Some of these constraints can be mitigated by coupling two or three rings with different resonance frequencies, as proposed in Ref 53. The number of lattice sites that can be coupled along the synthetic frequency dimension is ultimately limited by group velocity dispersion (GVD). For the fiber-based setup considered here, the GVD is 23 ps²/km at telecom wavelengths. We estimate a very small FSR drift of 7 mHz per mode of the resonator. This small value is due to the long resonator length. Thus $\sim 10^6$ modes are equally spaced to within a fraction of the 300 kHz cavity linewidth. Practically, the bandwidth of the fiber components in the cavity, especially the bandpass filter and the SOA gain, will limit the number modes to ~ 2000 .

In the revision, we add a Supplementary Discussion titled “**Extension to higher dimensional lattices**” where we elaborate on the above experimental challenges and approaches to mitigate them. We also add a sentence to the main text Discussion summarizing this: “**The dimensionality can be increased beyond 1D by using real-space dimensions [8, 9], by using additional frequency dimensions [34, 53], by using Floquet dimensions [22, 60] or by using other synthetic dimensions such as OAM in conjunction with frequency [66] (see Supplementary Discussion).**”

Reviewer #2 (Remarks to the Author):

In their manuscript “Experimental band structure spectroscopy along a synthetic dimension,” Dutt et al. theoretically proposed and experimentally demonstrated a technique to directly measure the band structure of a system with a synthetic dimension. To this end, they implemented a fiber ring resonator with a modulator that allows for independent tuning of the strength and range of the coupling along a synthetic lattice.

This is a very nice paper, I like it a lot. The idea is clearly presented, the experiments are convincing, the told story is smooth and interesting. I searched for things to criticize in the manuscript, I did not find any. There may be one optional thing: They authors might want to cite a recent arXiv about synthetic dimensions (arXiv:1903.07883); but this is completely up to the authors.

I recommend publication of the manuscript in Nature Communications.

We highly appreciate the reviewer’s favorable recommendation on our manuscript and for pointing us to the arXiv preprint on synthetic dimensions. In the revised version, we have cited the preprint [Ref 59], as well as other recent preprints and papers on synthetic dimensions [Refs 60, 64, 65, 74].

Reviewer #3 (Remarks to the Author):

In this manuscript, the authors present the paradigm of generating a synthetic dimension using parametric coupling between the free-spectral range states of a ring resonator. The scheme is, then, successfully realized experimentally. The field of generating synthetic dimensions for quantum simulation is nowadays extremely active. In this context, the authors provide an important but technical stepping stone towards the realization of more interesting physics. At the same time, the authors present a nice comprehensive theoretical overview of the Floquet analysis for the detection of the synthetic band spectrum. Combining all of this together, I however do not see the added merit of publishing this work in Nat. Comm. due to the following reasons:

We thank the reviewer for the comments recognizing the important advance produced by our work in a field that, as he/she states, is “nowadays extremely active.”

1. The theoretical analysis is not new:

a. The proposal of using this system for synthetic dimensions has already been proposed by some of the authors of this paper, as well as in other works [9, 39, Nature Physics 13, 545 (2017)].

We agree with the reviewer that the theoretical proposal of realizing synthetic frequency dimensions using modulated rings is not new. This is evidenced by our extensive reference to recent theoretical papers on synthetic frequency dimensions using modulated rings [8, 9, 26, 27, 37, 45, 53]. We have also added a citation to the Nature Physics 2017 article pointed out by the

reviewer [Ref 73]. However, the main claims of novelty in our work, as summarized in the abstract, are:

- (i) the theoretical proposal of directly reading out synthetic-space band structures,
- (ii) an experimental demonstration of this proposal,
- (iii) an experimental realization of the synthetic frequency dimension using a modulated ring,
- (iv) a demonstration of the reconfigurability of the lattice by tuning the amplitude and frequency of the modulation, enabling band structure engineering, long-range hopping, gauge potentials and nonreciprocal bands.

Note that Ref 39 of the original manuscript (Ref 40 in the revision), and the Nature Physics 2017 article [Ref 73], realize a synthetic *time* dimension using pulses in fiber loops, different from our modulated fiber ring that realizes a *frequency* dimension using a continuous-wave input. We have already mentioned this in the Discussion section.

b. The input-output formalism for detection of photonic bands has been around for some time, see e.g., [Phys. Rev. A 84 043804 (2011), Phys. Rev. Lett. 112 133902 (2014)].

The references alluded to by the reviewer deal with the detection of *bands* for *real-space systems* as opposed to our work on the detection of the *band structure* in *synthetic space*. We make an important distinction here between “*bands*” and “*band structure*”. The former refers to the frequency ranges which are allowed to propagate in the system, separated by band gaps. This is more akin to the density of states, as shown in Fig. 3(g)-(i) of our paper (see also Fig. 3 of the suggested PRA 2011 paper by Umucalilar and Carusotto). On the contrary, a band structure refers to the explicit dependence of the eigenenergies ϵ_k on the Bloch wavevector or quasimomentum k [see Fig. 3(a)-(f) and Fig. 4 of our paper], which provides much more information than the density of states. The density of states can be obtained from the band structure by integrating out the k -dependent information, e.g. using the Green’s function. The input-output formalism [PRA 2011] that the reviewer alludes to provides a way of detecting photonic bands, and was for real-space systems. In the synthetic frequency dimension, the Bloch wavevector k is identical with the time axis, hence it enables a unique way to detect the band structure, not only the photonic bands. Our manuscript introduces this method for the first time, and theoretically establishes this method of measuring the band structure using a rigorous Floquet analysis.

To this end, we add a sentence to the introduction: “This is in contrast to previous proposals of detecting density of states in real-space systems [PRA 2011, PRL 2014]. These proposals do not directly reveal the k -dependence of eigenenergies, and hence do not provide a direct detection of the band structure.”

c. The time analysis for reading of Floquet bands is also widely used, for example, [Phys. Rev. X 4, 031027 (2014), Phys. Rev. A 96, 053602 (2017)].

Although Floquet analysis is widely used in previous work, the mapping of the time axis to the reciprocal space wavevector k is established for the first time in our work. To establish this, we use the input-output formalism and Floquet analysis. In the revised manuscript, we include citations to the PRX 2014 and PRA 2017 papers mentioned by the reviewer. Here we reiterate the discussion after Eq. (13) that is unique to our manuscript: “...for a fixed input detuning $\Delta\omega$ that is within a band of the system, the temporally-resolved transmission exhibits peaks at those times t for which the system has an eigenstate with $\epsilon_k = \Delta\omega, k = t$. Thus, measuring the times at which

the transmission peaks appear in each modulation period $2\pi/\Omega$, as a function of $\Delta\omega$, yields the Floquet band structure of the system.” To the best of our knowledge, this concept has not been predicted before.

2. Recently, there have appeared more advanced results on photonic realization of synthetic dimensions [Nature 567, pages356–360 (2019)].

Our paper already mentions this article from the Segev group [Ref 12 in original manuscript], which presents an excellent realization of 1 real spatial dimension and 1 synthetic spatial dimension using supermodes. We believe our experimental demonstration nicely complements such efforts in realizing photonic synthetic dimensions. We reiterate that despite the central theoretical significance of a band structure in describing synthetic lattices, all previous work in synthetic dimensions – both for photons and ultracold atoms – have not provided a direct experimental measurement of the band structure. This is also applicable to the recent Nature 2019 paper noted by the reviewer. Our work realizes a synthetic frequency dimension and directly measures its band structure. This is facilitated by the role played by the time axis as the reciprocal space of the frequency dimension – a role that is unique to the synthetic frequency lattice. Additionally, we demonstrated complex long-range coupling along the frequency dimension, whereas Ref 12 only had nearest-neighbor coupling. Finally, in comparison to the synthetic spatial dimension implementation in Ref 12, which is restricted to 2D since the z-axis is used to simulate propagation in time, the frequency dimension holds promise for augmenting spatial dimensions to realize a 4D lattice, as discussed by Ozawa et al. [Ref 9], or an arbitrary higher-dimensional lattice using very few rings, as discussed by our group [Ref 53]. We discuss such extensions to higher dimensional lattices in detail in the Supplementary Discussion.

REVIEWERS' COMMENTS:

Reviewer #1 (Remarks to the Author):

The authors have addressed all my comments. The paper can be published in its current form.

Reviewer #1 (Remarks to the Author):

The authors have addressed all my comments. The paper can be published in its current form.

Response to Reviewer #1:

We thank the reviewer for recommending the publication of our manuscript.